# Effects of Dietary Sunflower Hulls on Performance and Rumen Fermentation of Pregnant Naemi Ewes: A Sustainable Fiber Source for Arid Regions

**DOI:** 10.3390/ani15243569

**Published:** 2025-12-11

**Authors:** Mohsen M. Alobre, Ibrahim A. Alhidary, Mohammed M. Qaid, Abdulrahman S. Alharthi, Ahmad A. Aboragah, Riyadh S. Aljumaah, Mutassim M. Abdelrahman

**Affiliations:** Department of Animal Production, College of Food and Agriculture Sciences, King Saud University, Riyadh 11451, Saudi Arabia; ialhidary@ksu.edu.sa (I.A.A.); abalharthi@ksu.edu.sa (A.S.A.); aaboragah@ksu.edu.sa (A.A.A.); rjumaah@ksu.edu.sa (R.S.A.); amutassim@ksu.edu.sa (M.M.A.)

**Keywords:** digestibility, Naemi ewes, rumen fermentation, serum biochemistry, sunflower hulls, total mixed ration

## Abstract

In many arid areas such as Saudi Arabia, good-quality forage for sheep is expensive or difficult to obtain, especially for pregnant ewes that require consistent nutrition. This study explored whether sunflower hulls (SFH)—a low-cost by-product that is widely available in the region—can be used as a reliable fiber source in ewe diets. We tested different amounts of SFH in the feed of pregnant Naemi ewes and monitored their eating behavior, body condition, rumen health, and basic blood indicators. Ewes fed higher levels of SFH ate slightly more feed, and their overall body weight remained stable throughout pregnancy. Most measures of nutrient digestion did not change, and the animals showed normal rumen activity and healthy blood values, suggesting the diets were safe and well tolerated. Feeding very high levels caused some shifts in rumen fermentation, so moderate inclusion was more suitable. Overall, the results show that SFH can be used to partially replace traditional forages in regions with limited resources. Including about 12–20% SFH in the diet provides a practical and affordable option to support pregnant ewes under arid feeding conditions.

## 1. Introduction

Feeding strategies that optimize nutrient utilization and minimize costs are critical for sustainable livestock production in arid and semi-arid environments such as those of the Arabian Peninsula [1,2]. Small ruminant productivity in these regions—such as central and eastern Saudi Arabia—is strongly limited by the scarcity of natural pastures, the rapid decline of rangeland biomass during the long dry season, and the heavy reliance on imported forages like alfalfa and Rhodes’s grass [3]. These constraints substantially increase feeding costs and expose producers to market and supply fluctuations. In response, many farms have adopted total mixed ration (TMR) systems, which help stabilize nutrient intake, improve feed efficiency, and reduce performance variability under harsh climatic conditions [3,4,5]. Within these systems, the use of agro-industrial by-products as alternative fiber sources has become increasingly attractive, offering a cost-effective and environmentally sustainable solution to the ongoing challenges of feed scarcity and rising production costs in desert livestock systems.

Recently, herbs or flowering plants such as flaxseed and sunflower seeds have gained recognition as fiber-rich foods. Flaxseed provides about 27–28 g of fiber per 100 g, while sunflower seeds offer around 22.6 g/100 g. Their high fiber content supports digestive health, helps regulate blood sugar and cholesterol, and promotes satiety [6,7,8,9]. In the oil extraction industry, substantial amounts of residues are generated, and these by-products can be utilized as protein-rich feed for ruminants, contributing to more sustainable production practices and advancing a circular-economy approach [10,11]. Sunflower hulls (SFH), a lignocellulosic by-product generated after oil extraction from sunflower seeds (*Helianthus annuus* L.), are produced in large quantities year-round in countries such as Saudi Arabia and neighboring Middle Eastern regions, where annual sunflower seed processing yields substantial volumes of hulls that are often underutilized [3]. Nutritionally, SFH contain high levels of neutral detergent fiber (NDF; 50–60%) and moderate crude protein (5–8%), but their elevated lignin content (18–22%) can limit ruminal degradation and energy release if included excessively in ruminant diets [12]. Despite their availability and low cost, SFH are typically used only sparingly in commercial pelleted feeds due to concerns about reduced digestibility at high inclusion rates. However, limited research—particularly in Naemi ewes, a dominant breed adapted to arid Saudi conditions—has evaluated their practical feeding value, optimal inclusion level, or metabolic implications. This gap underscores the need to assess SFH within region-specific production systems to determine how this abundant by-product can be converted into a consistent and sustainable fiber resource.

Nevertheless, when properly processed via pelleting or grinding, SFH can enhance rumen function and reduce feed costs [13]. Previous studies on lambs and dairy ewes have shown variable responses in digestibility, intake, and milk composition when plant residues are used as major fiber sources [14,15]. However, data regarding the safe inclusion rate of SFH in pregnant Naemi ewes, a major indigenous breed in Saudi Arabia, remain limited. Pregnancy and lactation are physiologically demanding stages during which nutrient partitioning changes rapidly. Fluctuations in dry matter intake, body condition score (BCS), and serum metabolites such as glucose, total protein, and triglycerides are expected [16,17]. Therefore, dietary fiber level and quality may directly affect rumen fermentation dynamics, total volatile fatty acids (VFAs) production, and overall energy supply during these periods [18]. The present study was designed to determine the effect of graded levels of SFH (0%, 12%, 20%, and 28%) in pelleted complete diets on the productive performance, apparent digestibility, ruminal fermentation pattern, and serum biochemical profile of pregnant Naemi ewes during different physiological stages. It was hypothesized that moderate inclusion of SFH would improve fiber intake and maintain normal metabolic activity without compromising digestibility or rumen fermentation efficiency.

## 2. Materials and Methods

### 2.1. Ethical Approval

The study was carried out at the Experimental Station of the Department of Animal Production, Faculty of Food and Agricultural Sciences, King Saud University, Riyadh, Saudi Arabia and Scientific Research Ethics Committee guidelines were followed for all the research projects (approval number: KSU-SE-20-27).

### 2.2. Experimental Animals and Management

A total of 84 healthy multiparous Naemi ewes (average initial body weight = 14.41 ± 4.8 kg; BCS = 2.88 ± 0.15) were used in this study. Animals were purchased from the local livestock market, dewormed, vaccinated against clostridial diseases, ear-tagged, and adapted to the facilities and basal diet for 14 days prior to the start of the study. Animals were clinically examined prior to enrollment and were deemed free of metabolic or infectious disorders. To ensure balanced allocation across treatments, ewes were first stratified by body weight and BCS and then randomly assigned to one of four dietary treatments (*n* = 21 ewes per treatment). Each treatment consisted of seven replicate pens, with three ewes per pen. The ewes were housed in shaded pens (7.0 m × 6.0 m) equipped with individual feed troughs and water buckets. All animals had free access to fresh water throughout the study. Diets were offered twice daily (08:00 and 15:00 h) at approximately 2% of initial body weight. The trial was conducted during three physiological stages: early gestation, late gestation, and early lactation, and lasted 240 days.

### 2.3. Diets, Sunflower Hulls Source, and TMR Preparation

SFH was sourced from a sunflower oil–processing facility in Riyadh, Saudi Arabia, where sunflower seed processing occurs year-round, ensuring a consistent supply. The SFH were air-dried, screened to remove foreign material, and ground to a 2–3 mm particle size prior to incorporation into total mixed rations (TMR).

The TMRs provided an average of 3.67 Mcal GE kg^−1^ and 14.6% crude protein. Four pelleted TMR diets were formulated to contain 0% (control), 12%, 20%, or 28% SFH on a dry matter basis. Diets were formulated to meet or exceed NRC [19] nutrient requirements for pregnant ewes. The TMR ingredients were mixed using a horizontal mixer (Model HM-100, Zhengzhou AG Machinery Co., Ltd., Zhengzhou, China), steam-conditioned to 70–75 °C, and pelleted through a 4-mm die. Pellets were cooled to ambient temperature before feeding. The ingredient and nutrient composition of each diet are presented in Table 1, as well as the composition of SFH based on NRC [19], as presented in Table 1.

### 2.4. Feed Sampling and Chemical Analysis

Representative samples of each dietary treatment were collected before the trial, ground to a 1-mm particle size using a Wiley mill (Thomas Scientific, Swedesboro, NJ, USA), stored at −20 °C, and later composited for proximate analysis.

Dry matter intake (DMI) was determined by oven-drying at 100 °C for 24 h. Ash content was determined by incineration at 550 °C for 3 h in a muffle furnace. Crude protein (CP) was analyzed using the Kjeldahl method (AOAC 2005; Method 984.13) as described in Bonanno et al. [20]. Acid detergent fiber (ADF) and NDF were determined following the procedures of Van Soest et al. [21], detergent system using heat-stable α-amylase and sodium sulfite where appropriate (AOAC 2005; Method 2002.04) [20]. Acid detergent lignin (ADL) was quantified using sulfuric acid digestion.

### 2.5. Feed Intake and Body Weight Measurement

Feed offered and refused was recorded weekly to calculate DMI for each treatment group. Ewes were weighed before morning feeding (07:30 h) at baseline and every four weeks thereafter using an electronic small-animal scale (Model EBSC-150, Adam Equipment Co., Milton Keynes, UK). Body weight and feed intake data were expressed as means for each physiological stage (growth, early gestation, late gestation, and lactation).

### 2.6. Body Condition Scores

It was used to assess the nutritional status of the ewes during late gestation (−30 d prepartum) and early lactation (+30 d postpartum). Forty-eight ewes (12 per treatment) were evaluated using the dorsal-palpation technique described by Santucci and Maestrini [22], on a 0–5 scale with 0.5 increments. Animals were grouped into four BCS classes (≤2.0, 2.5, 3.0, ≥3.5).

### 2.7. Digestibility Trial and Analysis

A digestibility sub-trial was conducted using eight ewes per treatment (Total 32 Naemi ewes in late pregnancy). Animals were moved to individual metabolism crates (1.5 m × 0.70 m) designed for complete fecal and urine collection. Following a 7-day adaptation period, feces were collected for five consecutive days.

During collection, feed offered, refusals, and fecal output were weighed daily. Representative samples (feed = 5%, feces = 20%) were stored at −20 °C until analysis. Dried and ground (1 mm) samples were proximately analyzed for NDF, ADF, cellulose, hemicellulose, and lignin [20]. Apparent nutrient digestibility coefficients were calculated using standard procedures.Apparent digestibility (%) = [(Intake − Fecal Output)/Intake] × 100.

### 2.8. Rumen Fermentation Profile

Rumen fluid was collected from nine ewes per treatment 30 days pre- and postpartum via stomach tubing before morning feeding and three hours after feeding. Samples were immediately filtered through four layers of cheesecloth, and subsamples were acidified with 25% metaphosphoric acid (4:1 ratio) and stored at −20 °C. Two milliliters of 1 M H_2_SO_4_ were added to each aliquot to stop microbial activity. The pH was immediately measured with a microprocessor pH meter (Model pH 211; Hanna Instruments, Woonsocket, RI, USA).

Concentrations of total VFAs, acetate, propionate, butyrate, and valerate were analyzed using gas chromatography (Agilent 7890B GC system, Technologies Inc., Wilmington, DE, USA) equipped with a flame ionization detector (FID; FID; Agilent Technologies Inc., Santa Clara, CA, USA) and a DB-FFAP silica capillary column (30 m × 0.25 mm × 0.25 µm; Agilent Technologies Inc., Santa Clara, CA, USA). The injector and detector temperatures were set at 220 °C and 250 °C, respectively. The oven temperature program ranged from 110 °C to 180 °C at a rate of 10 °C/min. Nitrogen was used as the carrier gas [23,24].

### 2.9. Blood Collection and Biochemical Analysis

Monthly blood samples (10 mL each) were obtained from the jugular vein of ewes using 10-mL vacutainer tubes before the morning feeding (07:00 h). Serum was separated by centrifugation at 2400× *g* for 15 min at 4 °C and stored at −20 °C. Serum metabolites-including glucose, total protein, albumin, urea-N, total cholesterol, and triglyceride- were analyzed using a semi-automated biochemical analyzer (RX Monza; Randox Laboratories, Crumlin, UK) with commercially available diagnostic kits.

### 2.10. Statistical Analysis

Data were collected using a completely randomized design with repeated measurements and analyzed using SAS [25], v. 9.4, SAS Institute Inc., Cary, NC, USA. All variables were tested for normality (Shapiro–Wilk test) and homogeneity of variance (Levene’s test) prior to analysis. A mixed model (PROC MIXED) was used with dietary treatment, physiological stage, and their interaction as fixed effects, and pen as a random effect. The resulting model for the performance, blood metabolites, and rumen fermentation of ewes was as follows:Y_ijk_ = μ + τ_i_ + T_j_ + τT_ij_ + ε_ijk_
where μ is the experimental mean, τi is the treatment effect, Tj is the time effect, τTij is the interaction between treatment and time, εij is the expected error, i = 1, 2, …, t, and j = 1, 2, …, r. The effects of dietary treatment (SFH concentration), measurement time, and their interaction (treatment × time) were tested.

To evaluate dose–response effects of increasing SFH inclusion, linear, quadratic, and cubic trend analyses were conducted using orthogonal polynomial contrasts.

The resulting model for the digestibility trial of ewes was as follows:Y_ij_ = μ + τ_i_ + ε_ij_
where τi is the treatment effect, εij is the expected error, i = 1, 2, …, t.

Results were expressed as least squares means ± SEM, and statistical significance was declared at *p* ≤ 0.05.

## 3. Results

### 3.1. Ewes’ Performance and Body Condition Scores

The effects of dietary SFH inclusion on the productive performance of ewes are presented in Table 2 and Figure 1. Dry matter intake responded clearly to increasing dietary SFH levels (*p* < 0.05). Ewes receiving the highest inclusion (28% SFH) consumed approximately 10% more feed than those on the control diet (1.47 vs. 1.34 kg d^−1^), indicating that the bulky but palatable nature of SFH stimulated intake rather than restricting it. Physiological stage also influenced intake patterns, with the lowest DMI observed during late gestation (1.27 kg d^−1^), when rumen capacity is naturally reduced by fetal growth, and the highest during early gestation (1.48 kg d^−1^). Both quadratic and cubic trends across SFH levels suggest a non-linear response in intake, with moderate inclusion levels (12–20%) producing slightly more stable intake patterns across stages.

Despite differences in DMI, body weight remained consistent among treatments (overall mean 52.67 kg; *p* > 0.05), demonstrating that increased intake at higher SFH levels did not translate into excessive weight gain or loss. BCS, however, was more sensitive to diet and stage (*p* < 0.05). During late gestation, control ewes maintained the highest BCS (3.40), while ewes fed 20% SFH showed a moderate reduction (3.00). Interestingly, during lactation—when energy demands increase—ewes fed 28% SFH displayed slightly improved BCS (2.90), suggesting better recovery of body reserves when higher fiber diets supported greater feed intake.

### 3.2. Digestibility Trial

Digestibility coefficients are summarized in Figure 2. Although dry matter and major fiber fractions (NDF and ADF) did not differ significantly among treatments (*p* > 0.05), meaningful patterns emerged with increasing SFH. Ewes fed the 28% SFH diet exhibited nearly double the apparent digestibility of ADL (50%) compared with the other groups, indicating a marked adaptive response of rumen microbes to the lignin-rich substrate. Numerical increases in NDF and ADF digestibility in the 28% SFH group further support enhanced microbial engagement with more recalcitrant fiber fractions, even if these differences did not reach statistical significance. Taken together, these findings suggest that moderate-to-high SFH inclusion may promote gradual microbial adaptation to lignocellulosic material.

### 3.3. Rumen Fermentation Pattern of Naemi Ewes

Rumen fermentation parameters are shown in Table 3. During late gestation (30 d prepartum), rumen pH increased with higher SFH inclusion (*p* = 0.01), reaching 6.90 in ewes fed 28% SFH, compared to 6.38 in controls, reflecting the buffering effect of fiber-rich diets. In contrast, total VFAs-an important indicator of fermentative energy yield-showed a marked linear decline (*p* = 0.01), decreasing from 114.4 mM in the control group to 57.5 mM in ewes fed 28% SFH, representing nearly a 50% reduction. This reduction was accompanied by parallel declines in acetate, propionate, butyrate, and valerate (*p* < 0.05), indicating a general shift toward lower fermentation intensity at higher SFH levels. Postpartum measurements showed similar patterns, with VFAs remaining lower in all SFH-containing diets (*p* < 0.01) despite no treatment effect on pH (*p* = 0.70). Across both sampling periods, the acetate:propionate (A:P) ratio increased quadratically with SFH inclusion, reaching the highest value in ewes fed 28% SFH, indicative of a more acetate-driven fermentation typical of high-fiber diets. Collectively, these results suggest that moderate inclusion (12–20%) preserved fermentation efficiency better than the highest level, which tended to depress total VFA production.

### 3.4. Serum Metabolic Profile of Ewes

Serum biochemical responses are reported in Table 4. Serum glucose did not differ significantly among treatments (*p* > 0.05). However, during early growth, ewes fed 28% SFH showed higher glucose concentrations (53.4 mg dL^−1^) than controls (38.6 mg dL^−1^), possibly reflecting enhanced glucose turnover associated with increased feed intake. During gestation, glucose levels declined linearly with rising SFH levels (*p* < 0.05), consistent with greater fetal glucose uptake in high-litter-demand periods.

Total protein concentrations varied by treatment and stage (*p* < 0.05). During pregnancy, control ewes exhibited slightly higher protein levels. However, during lactation, the 12% and 20% SFH groups showed better protein status than the 28% SFH group, indicating more efficient dietary protein utilization at moderate SFH inclusion levels. Serum albumin increased linearly with SFH inclusion during the growth stage (*p* < 0.05), with the highest values in ewes fed 28% SFH (4.66 g dL^−1^).

Lipid metabolites were sensitive to diet. Cholesterol and triglyceride concentrations were highest in the 12% and 20% SFH groups during lactation (*p* < 0.01), whereas the 28% SFH group consistently exhibited lower levels. This pattern suggests reduced lipid mobilization at the highest SFH inclusion level, likely due to greater fiber intake and a lower fermentative energy yield. Serum urea-N was not affected by diet overall (*p* > 0.05), though it increased linearly during gestation with higher SFH levels (*p* < 0.001), potentially indicating increased protein turnover associated with elevated DMI in late pregnancy. Notably, all serum metabolites remained within the physiological range for healthy ewes, confirming that dietary SFH up to 28% did not compromise systemic metabolic function.

## 4. Discussion

The present study examined the impact of increasing levels of SFH in pelleted TMR on feed intake, digestibility, ruminal fermentation, and blood biochemical parameters of pregnant and lactating Naemi ewes. The results indicate that SFH is a viable alternative fiber source in arid-region feeding systems, provided that inclusion levels are optimized to maintain an adequate fermentative energy supply.

The progressive increase in DMI with higher SFH inclusion, particularly in the 28% SFH group, suggests that the bulky, fibrous nature of SFH stimulated rumination and rumen fill without compromising palatability. Similar responses have been reported when sunflower-based by-products were incorporated into small-ruminant diets, demonstrating that animals in arid zones adapt well to fibrous residues when provided in pelleted form [14,26]. The slight improvement in BCS during lactation in ewes fed 28% SFH may reflect increased rumen fill and fiber-driven satiety rather than enhanced energy retention, consistent with findings from Mahgoub et al. [27] and Zhang et al. [28] showing that moderate-fiber pelleted diets stabilize body reserves in desert-adapted sheep.

Although total DM and major fiber fraction digestibilities were not significantly altered, the marked increase in ADL digestibility at 28% SFH suggests a degree of microbial adaptation to lignocellulosic substrates. Pelleting and fine grinding (1-mm particle size) likely improved physical accessibility, reduced particle size, and promoted microbial colonization, as previously noted by Osman et al. [13] and Tsvetanova et al. [29] for sunflower residues. Numerical increases in NDF and ADF digestibility at higher SFH levels also imply enhanced microbial hydrolysis efficiency despite the high lignin content. These findings align with reports showing improved pH stability and fiber utilization when pelleted rations supply adequate physically effective fiber [19].

Rumen fermentation was sensitive to the graded inclusion of SFH. The reduction in total VFAs and individual acids at higher SFH levels reflects the lower fermentability of lignin-bound carbohydrates, which slows microbial hydrolysis and limits substrate availability. This response is consistent with studies demonstrating that diets rich in recalcitrant fiber or high forage-to-concentrate ratios yield reduced VFA production due to restricted access of rumen microbes to structural carbohydrates [3]. Despite being a high-fiber ingredient, SFH did not increase acetate concentration—in fact, both acetate and propionate declined at higher inclusion levels. This indicates that the limiting factor was not fermentation pathway preference but the overall fermentable energy available for microbial metabolism [20]. As SFH contains considerable lignin (18–22%), a portion of the fiber may have remained resistant to enzymatic breakdown, thereby reducing total substrate conversion and overall VFAs output [30,31]. The quadratic rise in the acetate:propionate ratio, therefore, reflects a proportional shift within a reduced total VFAs pool rather than an actual increase in acetate production. Consequently, although diets with 28% SFH were able to maintain rumen pH within normal physiological limits, they exhibited lower fermentation efficiency and reduced energy yield per unit of DMI. These trends support the concept that moderate SFH inclusion (12–20%) provides sufficient effective fiber without excessively restricting fermentable carbohydrate availability.

Serum biochemical values remained within physiological limits for healthy ewes [32], demonstrating that SFH inclusion did not disrupt metabolic homeostasis. The linear decline in serum glucose during gestation at higher SFH levels likely reflects increased fetal glucose demand, consistent with earlier reports by Balıkcı, E., A. Yıldız and F. Gürdoğan [33] and Pesántez-Pacheco et al. [34]. Higher glucose during growth and lactation in ewes fed 28% SFH may be related to their elevated DMI, which increased gluconeogenic precursor supply. Total protein and albumin variations across treatments appear to reflect changes in dietary protein utilization and microbial protein synthesis. Ewes receiving 12–20% SFH maintained more favorable protein profiles during lactation, suggesting adequate rumen nitrogen synchrony at moderate fiber inclusion. These responses align with findings in Barki ewes under semi-arid conditions [35]. Lipid metabolites also supported the superiority of moderate SFH inclusion. Higher serum cholesterol and triglycerides in the 12% and 20% SFH groups indicate active lipid mobilization to support milk synthesis, whereas the lower levels observed at 28% inclusion likely reflect reduced dietary energy density. This pattern is consistent with trends reported in high-fiber sheep diets by Zhang et al. [28]. Serum urea-N concentrations increased linearly with SFH level during gestation, likely due to higher DMI and increased protein turnover; however, all values remained within normal ranges [36], confirming adequate protein balance in all diets.

Collectively, the performance, rumen fermentation, and biochemical data point to a coherent biological response: moderate SFH inclusion (12–20%) supports optimal intake, maintains adequate fermentative energy production, and preserves metabolic stability. In contrast, although 28% SFH is physiologically safe, its high lignin content limits hydrolysis efficiency and reduces total VFA production, which may constrain energy availability during late gestation and early lactation. These results demonstrate that Naemi ewes are capable of adapting to pelleted TMR containing lignocellulosic by-products such as SFH, particularly at moderate inclusion levels. Incorporating SFH into sheep diets in arid environments contributes to circular bioeconomy practices by converting abundant agro-industrial residues into valuable feed resources while reducing reliance on imported forages. Based on the mechanistic evidence and cross-parameter integration observed in this study, an inclusion level of 12–20% SFH is recommended for practical feeding programs, offering an effective balance between cost, digestibility, and rumen fermentation efficiency.

## 5. Conclusions

In conclusion, this study demonstrates that SFH can be used as a practical and sustainable fiber source for Naemi ewes in arid production systems where conventional forages are limited or costly. Based on the overall nutritional response and rumen fermentation trends, a moderate inclusion level of 12–20% SFH is recommended, as it supports stable physiological performance while maintaining fiber effectiveness and diet affordability. Incorporating SFH into pelleted TMRs offers producers a viable strategy to reduce dependence on imported forages and enhance feed resource resilience in desert environments. Future research should explore the long-term implications of SFH feeding on milk production, rumen microbial adaptation, and metabolic efficiency, as well as its integration into broader alternative-fiber strategies aimed at strengthening the sustainability of small ruminant systems under harsh climatic conditions.

## Figures and Tables

**Figure 1 animals-15-03569-f001:**
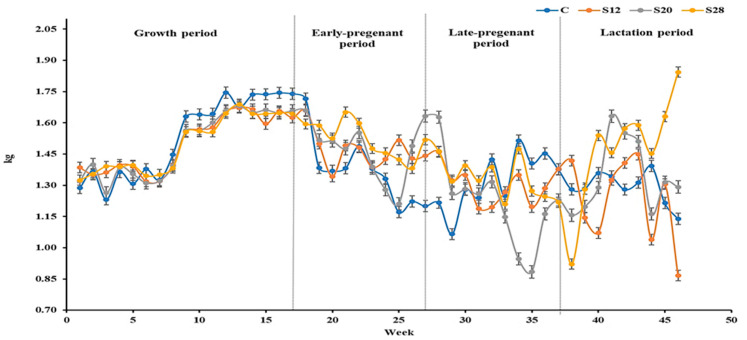
Mean dry matter intake (kg/d) of Naemi ewes during the different experimental periods. *n* = 21; Treatments C, S12, S20, and S28 represent complete pelleted diets supplemented with 0%, 12%, 20%, and 28% sunflower hulls, respectively.

**Figure 2 animals-15-03569-f002:**
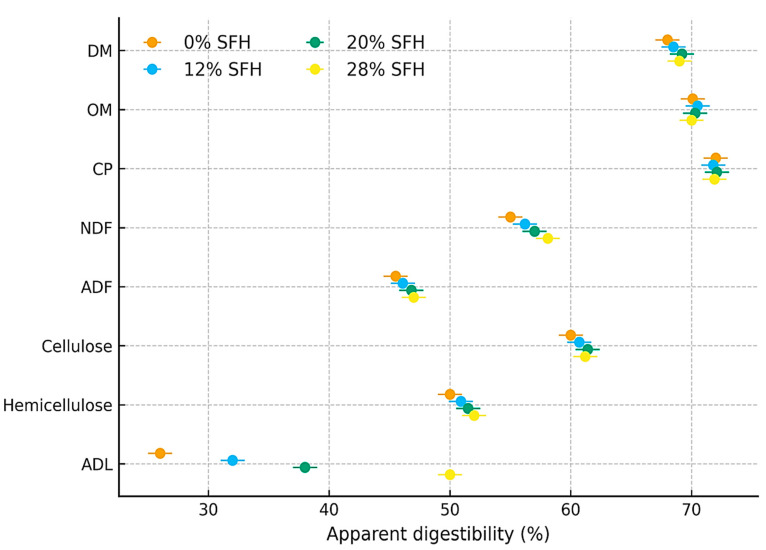
Apparent digestibility coefficient of dry matter and fiber by Naemi ewes fed different experimental diets. Treatments: Complete pelleted diets supplemented with 0%, 12%, 20%, or 28% sunflower hulls (SFH); DM, dry matter; OM, organic matter; CP, crud protein; NDF, neutral detergent fiber; ADF, acid detergent fiber; ADL, acid detergent lignin.

**Table 1 animals-15-03569-t001:** Ingredient and chemical composition of the experimental diets and sunflower hulls.

Item	Sunflower Hull Level Treatments ^1^			
	0%	12%	20%	28%
Ingredients (% of dietary dry matter)				
Barley grain	32.56	20.41	22.21	29.20
Palm kernel meal	20.00	20.00	20.00	9.25
Wheat straw	16.30	16.33	6.51	0.00
Sunflower meal	13.80	13.84	13.71	16.03
Sunflower hulls	0.00	12.00	20.00	28.00
Wheat bran	10.00	10.00	10.00	10.00
Molasses	5.00	5.00	5.00	5.00
Acid buffer	0.8	0.8	0.8	0.8
Limestone	0.72	0.74	0.87	0.90
Salt	0.52	0.56	0.57	0.49
Vitamin and mineral premix	0.30	0.30	0.30	0.30
Total	100.0	100.0	100.0	100.0
Chemical composition, % dry matter basis				
Dry matter	90.39	88.47	88.74	88.54
Protein	14.86	14.55	14.18	14.98
Fiber	18.26	20.78	22.16	21.81
Ash	14.25	6.61	6.34	5.88
Fat	4.02	4.35	4.35	4.00
Salt	0.80	0.80	0.80	0.80
ADF	28.46	30.25	30.66	29.55
NDF	36.50	37.59	39.74	41.52
Lignin	7.37	6.99	8.88	9.07
Cellulose	19.77	21.35	18.51	21.06
Hemicellulose	6.96	8.49	9.13	10.87
Gross energy (Cal/g)	3641	3613	3710	3744
Nutritive values of SFH based on NRC				
Chemical composition,% dry matter basis		Sunflower hulls (%)		
Dry matter		90.00		
Crude Protein		4.00		
Crude fiber		52.00		
NDF		73.00		
ADF		63.45		
Acid detergent lignin		22.01		
Ether extract		2.20		
Ash		3.00		
Calcium		1.47		
Phosphorus		0.11		
ME (Mcal/kg)		1.4		

^1^ Treatments: Complete pelleted diets supplemented with 0%, 12%, 20%, or 28% sunflower hulls; NDF, neutral detergent fiber; ADF, acid detergent fiber.

**Table 2 animals-15-03569-t002:** Effects of different levels of sunflower hulls supplementation on productive performance of Naemi ewes fed complete pelleted diets.

Measurement, Unit	Sunflower Hull Level Treatments ^1^				SEM ^2^	*p* Value
	0%	12%	20%	28%		
Feed intake, kg/d						
Growth	1.51 ^a^	1.49 ^ab^	1.48 ^b^	1.49 ^ab^	0.01	0.05 ^C^
Early gestation	1.37 ^c^	1.48 ^b^	1.53 ^a^	1.54 ^a^	0.03	0.01 ^Q^
Late gestation	1.31 ^ab^	1.26 ^b^	1.18 ^c^	1.36 ^a^	0.01	0.01 ^Q^
Lactation	1.34 ^b^	1.17 ^c^	1.31 ^b^	1.50 ^a^	0.02	0.01 ^Q^
Overall	1.34 ^b^	1.35 ^b^	1.37 ^b^	1.47 ^a^	0.02	0.02 ^P, T*P^
Body weight, kg						
Growth	41.75	41.63	42.02	41.86	0.31	0.97
Early gestation	50.76	51.09	50.51	50.25	0.40	0.89
Late gestation	58.25	59.88	58.78	61.01	0.46	0.16
Lactation	59.91	58.14	57.38	59.23	0.87	0.76
Overall	52.67	52.67	52.67	52.67	0.51	0.69
Body condition score						
Late gestation	3.40 ^a^	3.31 ^a^	3.00 ^b^	3.16 ^b^	0.35	0.05 ^Q^
Lactation	2.75 ^ab^	2.68 ^ab^	2.31 ^c^	2.90 ^a^	0.16	0.04 ^Q^
Overall	3.07	3.00	2.65	2.83	0.08	0.12

^a–c^ Within a row, means without a common superscript letter differ significantly (*p* < 0.05). *n* = 21; ^1^ Treatments: Complete pelleted diets supplemented with 0%, 12%, 20%, or 28% sunflower hulls; ^2^ SEM = Standard error of the means. *p* value represents the effect of dietary treatment; P: the effect of period; T*P = interaction effect between treatment and period. Q and C indicate quadratic and cubic responses of dietary treatments, respectively.

**Table 3 animals-15-03569-t003:** Effects of different levels of sunflower hulls supplementation on ruminal fermentation profile of ewes fed complete pelleted diets.

Measurement, Unit	Sunflower Hull Level Treatments ^1^				SEM	*p* Value
	0%	12%	20%	28%		
pH value						
Prepartum, 30 d	6.38 ^b^	6.47 ^ab^	6.57 ^ab^	6.90 ^a^	0.28	0.01 ^L^
Postpartum, 30 d	6.06	6.12	6.06	6.07	0.35	0.70
Overall	6.22	6.30	6.32	6.49	0.05	0.86 ^P, T*P^
Total VFAs, mM						
Prepartum, 30 d	114.38 ^a^	94.94 ^ab^	75.50 ^bc^	57.53 ^c^	7.17	0.01 ^L^
Postpartum, 30 d	102.67 ^a^	58.47 ^b^	62.15 ^b^	67.32 ^b^	7.07	0.006 ^L^
Overall	108.53	76.71	68.83	62.43	5.00	0.001 ^P, T*P^
Acetate, mM						
Prepartum, 30 d	54.70 ^a^	46.15 ^a^	30.38 ^b^	28.75 ^b^	3.38	0.04 ^L^
Postpartum, 30 d	50.14 ^a^	27.21 ^b^	31.13 ^b^	35.17 ^ab^	3.56	0.001 ^L^
Overall	52.42	36.68	30.755	31.96	2.56	0.001
Propionate, mM						
Prepartum, 30 d	28.50 ^a^	24.15 ^ab^	22.68 ^b^	16.21 ^b^	2.06	0.05 ^L^
Postpartum, 30 d	31.55 ^a^	17.10 ^b^	16.64 ^b^	17.31 ^b^	1.97	0.002 ^L^
Overall	30.03	20.63	19.66	16.76	1.36	0.001 ^T*P^
Butyrate, mM						
Prepartum, 30 d	22.47 ^a^	20.48 ^ab^	13.97 ^b^	5.65 ^c^	1.83	0.01 ^L^
Postpartum, 30 d	17.82 ^a^	14.15 ^b^	12.50 ^ab^	11.22 ^ab^	1.72	0.05 ^L^
Overall	20.15	17.32	13.24	8.44	1.27	0.001
Valerate, mM						
Prepartum, 30 d	2.55 ^a^	2.17 ^ab^	1.87 ^cb^	1.36 ^c^	1.13	0.01 ^L^
Postpartum, 30 d	2.17 ^a^	1.29 ^b^	1.98 ^a^	1.24 ^b^	0.12	0.001
Overall	2.36	1.73	1.93	1.30	0.09	0.001 ^P, T*P^
Acetate: Propionate ratio						
Prepartum, 30 d	1.98 ^b^	2.07 ^b^	1.26 ^b^	1.77 ^a^	0.20	0.02 ^Q^
Postpartum, 30 d	1.56 ^ab^	1.59 ^b^	1.87 ^ab^	2.03 ^a^	0.15	0.05 ^Q^
Overall	1.77	1.83	1.26	1.87	0.12	0.03 ^P, T*P^

^a–c^ Within a row, means without a common superscript letter differ significantly (*p* < 0.05). *n* = 21. ^1^ Treatments: Complete pelleted diets supplemented with 0%, 12%, 20%, or 28% sunflower hulls. SEM = Standard error of the mean. *p*-value represents the effect of dietary treatment. P:effect of physiological period stage; ^T*P^: interaction effect between treatment and period. ^L^, ^Q^, and ^C^ indicate linear, quadratic and cubic responses of dietary treatments, respectively.

**Table 4 animals-15-03569-t004:** The effect of different levels of sunflower hulls supplementation on biochemical profile of Naemi ewes.

Measurement, Unit	Treatments Based on Sunflower Hull Levels ^1^				SEM ^2^	*p* Value
	0%	12%	20%	28%		
Glucose mg/dL						
Growth	38.55 ^b^	49.74 ^ab^	40.04 ^b^	53.38 ^a^	2.25	0.04 ^C^
Pregnant	48.40 ^a^	48.78 ^a^	46.32 ^ab^	33.73 ^b^	2.45	0.05 ^L^
Lactation	68.95	72.99	74.54	59.80	3.70	0.48
Overall	51.96	57.17	53.63	48.97	2.03	0.33 ^P^
Total protein g/dL						
Growth	6.58	6.59	6.32	6.52	0.09	0.70
Pregnant	6.75 ^a^	6.67 ^b^	6.71 ^ab^	6.33 ^d^	0.03	0.001 ^C^
Lactation	6.56 ^c^	7.01 ^a^	7.01 ^a^	6.92 ^b^	0.04	0.01 ^Q^
Overall	6.63 ^ab^	6.75 ^a^	6.68 ^ab^	6.59 ^b^	0.04	0.05 ^Q P, T*P^
Albumin g/dL						
Growth	3.00 ^b^	3.50 ^b^	4.28 ^a^	4.66 ^a^	0.23	0.04 ^L^
Pregnant	4.12	3.54	4.19	4.50	0.20	0.37
Lactation	3.90	3.36	2.80	2.98	0.20	0.18
Overall	3.67	3.47	3.75	4.04	0.13	0.38 ^P, T*P^
Total cholesterol mg/dL						
Growth	61.60	57.96	58.01	61.30	2.29	0.91
Pregnant	70.27 ^b^	65.05 ^c^	75.12 ^a^	74.86 ^ab^	0.79	0.001 ^C^
Lactation	88.50 ^b^	98.51 ^a^	98.30 ^a^	78.00 ^c^	1.63	0.01 ^Q^
Overall	73.45 ^ab^	73.84 ^ab^	77.21 ^a^	71.38 ^b^	1.71	0.05 ^Q P, T*P^
Triglyceride’s mg/dL						
Growth	49.70	49.65	45.00	46.48	1.53	0.64
Pregnant	53.75 ^d^	56.15 ^c^	58.47 ^b^	59.27 ^ab^	1.06	0.001 ^L^
Lactation	58.61 ^b^	60.45 ^a^	60.45 ^a^	49.99 ^c^	0.83	0.001 ^Q^
Overall	54.02 ^ab^	56.65 ^a^	54.79 ^ab^	51.91 ^b^	0.79	0.01 ^Q P, T*P^
Urea-N mg/dL						
Growth	24.35 ^a^	20.13 ^cb^	22.11 ^ab^	19.81 ^c^	0.95	0.03 ^C^
Pregnant	15.92 ^c^	16.18 ^b^	16.09 ^bc^	20.01 ^a^	0.33	0.001 ^L^
Lactation	21.13 ^b^	21.82 ^a^	21.82 ^a^	20.94 ^c^	0.07	0.01 ^Q^
Overall	20.52	19.52	20.00	20.25	0.42	0.63 ^P, T*P^

^a,b,c^ Within a row, means without a common superscript letter differ significantly (*p* < 0.05). *n* = 21; ^1^ Treatments: Complete pelleted diets supplemented with 0%, 12%, 20%, or 28% sunflower hulls. ^2^ SEM = Standard error of the mean. P: effect of physiological period stage; ^T*P^: interaction effect between treatment and period.). ^L^, ^Q^ and ^C^ indicate linear, quadratic and cubic responses of dietary treatments, respectively.

## Data Availability

The data that support the findings of this study are available on request from the corresponding author upon reasonable request.

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
