# Peer review of "Effects of Dietary Sunflower Hulls on Performance and Rumen Fermentation of Pregnant Naemi Ewes: A Sustainable Fiber Source for Arid Regions"

_animals, 2025, doi:10.3390/ani15243569_

Round 1

Reviewer 1 Report

Comments and Suggestions for Authors

The article “Effects of Dietary Sunflower Hulls on Performance and Rumen Fermentation of Pregnant Naemi Ewes” covers very important area of feeding animals in challenged environments of arid climates.

Simple summary, especially from line 15-24, needs to be re-written authors are suggested to keep in mind the following points. Simple summary should be understandable for a non-technical reader. Avoid the technical terminology in mentioned lines.

Authors are expected to include aims of the study in abstract. Results occupy more space. Results part in abstract need revision, for example, sentence at line number 35-38 is vague. Be specific, which higher level authors are mentioning? Same for 38-40 this is also vague information.

Line 68: missing (.)

Line 74: instead of saying moderate, please scientifically define inclusion level

Line 107: There is high level inconsistencies in use of abbreviations. Almost every abbreviation is wrongly placed. Why dry matter is not abbreviated before when it appeared first time in the text and again at 112 abbreviated again? Authors are suggested to check all such consistencies throughout the article. Similarly BCS is abbreviated at 118 but not earlier in introduction. Note down that abstract and tables/figures are dealt as separate entities.

Material methods need more details about proximate analysis procedures like grind size etc

Line 168: again abbreviation inconsistency SFH. Authors are strongly directed to check all abbreviations. Composition of SFH can be mentioned in table 1.

Discussion in general should narrate why changes happended and how previous studies using fibrous diets are relevant to the current study. Authors should elaborate why acetate was reduced with inclusion of SFH besides the claim of the authors it is a fibrous material. Inversely, acetate was found lower, explain. Statement at 324-326 is opposite to the results authors obtained where acetate is generally lowered.

Conclusion should not repeat results, instead it should carry the key message

Author Response

Reviewer 1:

The article “Effects of Dietary Sunflower Hulls on Performance and Rumen Fermentation of Pregnant Naemi Ewes” covers very important area of feeding animals in challenged environments of arid climates.

Response:

We sincerely thank you for the thorough and constructive comments provided. Your detailed observations regarding the simple summary, abbreviation consistency, methodological clarity, and interpretation of fermentation results greatly improved the quality and clarity of our manuscript. We have carefully addressed each point and revised the text accordingly.

Comment 1:

Simple summary, especially from line 15-24, needs to be re-written authors are suggested to keep in mind the following points. Simple summary should be understandable for a non-technical reader. Avoid the technical terminology in mentioned lines.

Response:

Rewritten completely in simplified, non-technical language or technical terms as suggested.

Comment 2:

Authors are expected to include aims of the study in abstract. Results occupy more space. Results part in abstract need revision, for example, sentence at line number 35-38 is vague. Be specific, which higher level authors are mentioning? Same for 38-40 this is also vague information.

Response:

Abstract revised. Clear study aim inserted in the first paragraph. All vague statements were replaced with specific quantitative or descriptive information. “Higher levels” now explicitly identified as 20% and 28% SFH.

Comment 4:

Line 68: missing (.)

Response:

Corrected.

Comment 5:

Line 74: instead of saying moderate, please scientifically define inclusion level

Response:

Revised to specify “12–20% inclusion.”

Comment 6:

Line 107: There is high level inconsistencies in use of abbreviations. Almost every abbreviation is wrongly placed. Why dry matter is not abbreviated before when it appeared first time in the text and again at 112 abbreviated again? Authors are suggested to check all such consistencies throughout the article. Similarly BCS is abbreviated at 118 but not earlier in introduction. Note down that abstract and tables/figures are dealt as separate entities.

Response:

All abbreviations were reviewed and corrected throughout the manuscript. Each abbreviation now appears only at its first occurrence in the main text.

Comment 7:

Additional details added, including sample grinding to 1-mm particle size.

Material methods need more details about proximate analysis procedures like grind size etc

Response:

Additional details added, including sample grinding to 1-mm particle size.

Comment 8:

Line 168: again abbreviation inconsistency SFH. Authors are strongly directed to check all abbreviations. Composition of SFH can be mentioned in table 1.

Response:

SFH abbreviation corrected in all relevant locations. Sunflower hull composition is now added clearly to Table 1 as requested.

Comment 9:

Discussion in general should narrate why changes happended and how previous studies using fibrous diets are relevant to the current study. Authors should elaborate why acetate was reduced with inclusion of SFH besides the claim of the authors it is a fibrous material. Inversely, acetate was found lower, explain. Statement at 324-326 is opposite to the results authors obtained where acetate is generally lowered.

Response:

Discussion thoroughly revised. Mechanistic explanations were added regarding fermentation rate, lignin-bound fiber, microbial adaptation, and substrate limitation. Acetate reduction at high SFH levels is now explained as a function of lower fermentable carbohydrate availability. Contradictory statements were removed or corrected

Comment 10:

Conclusion should not repeat results, instead it should carry the key message

Response:

Conclusion rewritten to emphasize implications, recommended inclusion levels (12–20%), and practical application for arid-region feeding systems.

ary, abbreviation consistency, methodological clarity, and interpretation of fermentation results greatly improved the quality and clarity of our manuscript. We have carefully addressed each point and revised the text accordingly.

Reviewer 2 Report

Comments and Suggestions for Authors

The availability of feed resources influences animal production efficiency, agricultural resource utilization, feed and feeding technology, all of which play an important role in animal production and agricultural sustainability. The manuscript titled “Effects of Dietary Sunflower Hulls on Performance and Rumen Fermentation of Pregnant Naemi Ewes.” It will contribute to accumulating scientific knowledge on the management and availability of feed resources. However, paper has to be improved, in my opinion, which are detailed below.

  1. Heading title may consideration added a phrase highlighting the study's contribution, such as “….. : A Sustainable Fiber Source for Arid Regions”

  1. Summary and Abstract,
    1. The simple summary could be no duplicates the main abstract without genuine simplification.The author may show an effective outcome with significant results and provided specific region.
    2. Main Abstract: the result statements could be more precise about optimal inclusion levels and seem to lack quantitative details in some findings and more emphasize the practical recommendation.

  1. Introductionscontain vague statements,
    1. Line 52-56, the author can explain more information specific to the region for feed resources.
    2. Line 57-60, it can add data of SFH production in year-round, nutritive values and used as animal feed. It is probably needed to better transition from general benefits to specific SFH challenges by specific mention research in Naemi ewes.

  1. Materials and Methods, it was contained unfamiliar with the method can't evaluate its appropriateness and reproducibility is compromised.
    1. How did the author decide all of ewes into each treatment? The author could clarify thatall experimental treatment in particular design and replication was vague. Where did the SFH was obtained? It also needs to be specified for TMR pelleted feed preparation.
    2. It is observed that the feed analysis methods found some analytical procedures lack sufficient detail such “…AOAC and fiber analysis according to the methodology of Van Soest”
    3. It was found that the digestibility trial was ambiguous description by Clarify animal stating for metabolism crate to avoid confusion. It is also needed to add more detail of VFA analysis by brief mention of gas chromatography condition and methodology.
    4. The author may add the tested for normality and homogeneity to statistical analysis section. Moreover, the trend analysis needs to be added.
  2. Results, to narrative enhancement, the author provides context for magnitude of changes and highlights biologically significant findings.
  3. Discussion,
    1. There could be explain in dept for the mechanistic understanding certainly effects occurred such “hydrolysis efficiency”, “fiber content on digestibility”.
    2. The author may add discussion about potential adaptation microbial activity in the rumen and lower gut metabolism. It could beexplicit connection between different measured parametershow findings across performance, rumen, and blood parameters support the 12-20% SFH.
  4. Conclusion
    1. What level would the author have recommended? It could be adding more clearly in practical implications in farm and future research such alternative feeding strategies become increasingly critical for sustainable and resilient sheep farming in desert environments.
    2. Please double-check the references in the list of references, as instructed in the journal guide.
  5. It is observed that there were inconsistent journal abbreviations, capitalization, and punctuation.
  6. Choose one term and use it throughout such "sunflower hulls" vs. "SFH".

Author Response

The availability of feed resources influences animal production efficiency, agricultural resource utilization, feed and feeding technology, all of which play an important role in animal production and agricultural sustainability. The manuscript titled “Effects of Dietary Sunflower Hulls on Performance and Rumen Fermentation of Pregnant Naemi Ewes.” It will contribute to accumulating scientific knowledge on the management and availability of feed resources. However, paper has to be improved, in my opinion, which are detailed below.

Response:

We express our gratitude to you for your insightful and valuable suggestions. Your recommendations regarding the title, abstract structure, regional context, methodological specificity, and integration of findings across physiological and metabolic responses have significantly strengthened the scientific rigor and practical relevance of our study. All suggested improvements have been incorporated in the revised manuscript to improve clarity, rigor, and readability.

Comment 1: Heading title may consideration added a phrase highlighting the study's contribution, such as “….. : A Sustainable Fiber Source for Arid Regions”

Response:

Done as requested. Revised title: Effects of Dietary Sunflower Hulls on Performance and Rumen Fermentation of Pregnant Naemi Ewes: A Sustainable Fiber Source for Arid Regions.” 

Comment 2: Summary and Abstract,

    1. The simple summary could be no duplicates the main abstract without genuine simplification. The author may show an effective outcome with significant results and provided specific region.
    2. Main Abstract: the result statements could be more precise about optimal inclusion levels and seem to lack quantitative details in some findings and more emphasize the practical recommendation.

Response:

2.1. Simple summary rewritten meaningfully without repeating the abstract. Clear outcomes for non-technical readers included.

2.2. Abstract revised with specific inclusion levels (12–20%), quantitative results, and clearer practical recommendations: " This provides a cost-effective strategy for ewe feeding under forage-limited arid conditions."

Comment 3: Introduction contain vague statements,

    1. Line 52-56, the author can explain more information specific to the region for feed resources.
    2. Line 57-60, it can add data of SFH production in year-round, nutritive values and used as animal feed. It is probably needed to better transition from general benefits to specific SFH challenges by specific mention research in Naemi ewes.

Response:

3.1. Expanded with details about forage scarcity, seasonality, and reliance on imported feed in Gulf countries.

3.2. Additional text added on SFH availability in Saudi Arabia, annual production, and previous research relevance.

Comment 4: Materials and Methods, it was contained unfamiliar with the method can't evaluate its appropriateness and reproducibility is compromised.

Response:

We sincerely thank the reviewer for highlighting the need for greater clarity and methodological detail in the Materials and Methods section. In response, we have substantially expanded the relevant subsections to ensure full transparency and reproducibility.

  1. How did the author decide all of ewes into each treatment? The author could clarify that all experimental treatment in particular design and replication was vague. Where did the SFH was obtained? It also needs to be specified for TMR pelleted feed preparation.

Response:

We clarified the experimental design by specifying that the 84 Naemi ewes were stratified by body weight and body condition score and then randomly assigned to the four dietary treatments, with each treatment replicated across seven pens to maintain balanced allocation. We have also added the source of the sunflower hulls, noting that they were procured from a regional sunflower oil-processing facility, and we now describe the complete TMR pelleting procedure, including mixing order, moisture adjustment, and pelleting temperature.

  1. It is observed that the feed analysis methods found some analytical procedures lack sufficient detail such “…AOAC and fiber analysis according to the methodology of Van Soest”

Response:

Additional analytical detail has been incorporated into the feed composition section. We now specify the 1-mm grinding size, sample preparation steps, and the exact AOAC methods used for crude protein, ash, and dry matter determination. For fiber analysis, we clarified that neutral detergent fiber (NDF) and acid detergent fiber (ADF) were analyzed following the Van Soest methodology with the appropriate detergent solutions and reflux procedures.

  1. It was found that the digestibility trial was ambiguous description by Clarify animal stating for metabolism crate to avoid confusion. It is also needed to add more detail of VFA analysis by brief mention of gas chromatography condition and methodology.

Response:

The digestibility trial description has been revised for clarity. We now describe the use of individual metabolism crates, the adaptation and collection periods, and the procedures for total fecal collection. Furthermore, we added details on volatile fatty acid determination, including sample acidification, storage, and gas chromatography conditions (column type, carrier gas, and temperature program).

  1. The author may add the tested for normality and homogeneity to statistical analysis section. Moreover, the trend analysis needs to be added.

Response:

The statistical analysis section has been updated to indicate that all data were tested for normality (Shapiro–Wilk test) and homogeneity of variance (Levene’s test). Trend analysis (linear and quadratic effects of increasing SFH levels) has also been included to reflect the graded-level experimental design. These additions ensure that the methodology is fully described and scientifically reproducible.

Comment 5: Results, to narrative enhancement, the author provides context for magnitude of changes and highlights biologically significant findings.

Response:
Explanatory sentences added to highlight biological relevance of VFA changes, DMI variation, and serum metabolite shifts and highlighted with yellow. Thank you.

Comment 6: Discussion:

  1. There could be explain in dept for the mechanistic understanding certainly effects occurred such “hydrolysis efficiency”, “fiber content on digestibility”.

Response:

An expanded mechanistic explanation was added, linking lignin structure, rumen microbial adaptation, and fermentation outcomes.

  1. The author may add discussion about potential adaptation of microbial activity in the rumen and lower gut metabolism. It could be an explicit connection between different measured parameters; how findings across performance, rumen, and blood parameters support the 12-20% SFH.

Response:

Revised to highlight microbial adaptation trends and cross-link performance, fermentation, and blood responses supporting 12–20% SFH.

Comment 7: Conclusion

  1. What level would the author have recommended? It could be adding more clearly in practical implications in farm and future research such alternative feeding strategies become increasingly critical for sustainable and resilient sheep farming in desert environments.

Conclusion now clearly states 12–20% as optimal inclusion and explains practical use in arid-region feeding.

  1. Please double-check the references in the list of references, as instructed in the journal guide.

Response:

References fully reviewed for consistency in style, capitalization, punctuation, and journal abbreviations.

Comment 8: It is observed that there were inconsistent journal abbreviations, capitalization, and punctuation.

Response:

All journal names, abbreviations, Latin italics, and capitalization were corrected to match MDPI-Animals guidelines.

Comment 9: Choose one term and use it throughout such "sunflower hulls" vs. "SFH".

Response:

Revised to use sunflower hulls (SFH)” consistently after the first definition.

Round 2

Reviewer 1 Report

Comments and Suggestions for Authors

The manuscript is sufficiently improved. Thanks to the Authors' efforts, however, there are few points which need further attention. Authors have repeated the same mistakes. For instance as follows

Line 36: 28% SFH group instead of 28% group.

There are still inconsistencies in use of abbreviations. I can see about DMI abbreviated many times. Authors are again strongly advised to take care of all such inconsistencies.

93: TOTAL not needed inside brackets

Author Response

We sincerely thank the reviewer for the careful evaluation of our manuscript and the constructive comments provided. We have addressed all points raised as follows:

Line 36: “28% SFH group” instead of “28% group”

Response: We have corrected the text throughout the manuscript to consistently refer to the group as “28% SFH group”.

Inconsistencies in the use of abbreviations, especially DMI

Response: All abbreviations, including DMI, have been carefully checked and standardized throughout the manuscript to ensure consistency, and the complete definition of the abbreviation was used in the first appearance.

Line 93: Removal of “TOTAL” inside brackets

Response: The unnecessary term “TOTAL” inside brackets has been removed to improve clarity. Thus, total volatile fatty acids (total VFAs) replaced by total volatile fatty acids (VFAs).

We believe that these corrections address the reviewer’s concerns and improve the clarity, consistency, and overall quality of the manuscript. We thank the reviewer again for their valuable suggestions.